# A RATE-DISTORTION THEORY
# OF ADVERSARIAL EXAMPLES

## ABSTRACT

The generalization ability of deep neural networks (DNNs) is intertwined with model complexity, robustness, and capacity. Through establishing an equivalence between a DNN and a noisy communication channel, we characterize generalization and fault tolerance for unbounded adversarial attacks in terms of information-theoretic quantities. Invoking rate-distortion theory, we suggest that excess capacity is a significant cause of vulnerability to adversarial examples.

## 1 INTRODUCTION

The mathematical concept of information, as introduced by Shannon (1948) in the context of communication over noisy channels, has revolutionized fields like psychology and neuroscience. Despite the fact that deep learning has been heavily influenced by these fields, information theory is not the established framework for its development. Built on mainly empirically derived design principles, modern neural networks achieve impressive results. However, a theoretical understanding as to when good performance can be expected is lacking (Zhang et al., 2017; Arpit et al., 2017).

In particular, the inability to characterize generalization imposes hard constraints on the responsible deployment of deep learning in performance-critical settings where fault tolerance is required. The challenge of obtaining such a characterization arises from the difficulty of interpreting (Bau et al., 2017) and quantifying the uncertainty of the predictions (Gal, 2016; Guo et al., 2017). Furthermore, the adversarial examples phenomenon (Biggio et al., 2013; Szegedy et al., 2014; Nguyen et al., 2015; Athalye et al., 2018) shows that solely maximizing test accuracy can result in potentially unsafe model behavior.

Many studies have attempted to understand generalization via model complexity and introduced various measures to quantify it. However, Neyshabur et al. (2017) concludes that a combination of several measures is required to explain some of the empirically observed phenomena. A more compact description of generalization was proposed by Tishby & Zaslavsky (2015), who analyzed the learning problem in terms of mutual information. This measure incorporates many of the previously proposed characteristics, effectively summarizing the interplay between the data, network architecture, and the optimization procedure. We adopt this view for the image classification problem and make the following contributions:

1. Identifying a DNN with a noisy communication channel we formalize the trade-off between a model's prediction accuracy and sensitivity to adversarial examples through rate-distortion theory. Practically speaking, this is the information bottleneck (IB) trade-off in the context of supervised deep learning for image classification. We attribute sensitivity to adversarial examples to excess capacity, i.e., complexity of input representations.

2. We explain the mechanisms by which explicit regularization strategies determine the model's capacity, drawing on an intuitive sphere packing argument for the Gaussian communication channel. Weight decay has a natural interpretation as a power constraint on the channel, while batch normalization acts as an information-theoretic "short-circuit" between adjacent layers.

3. We demonstrate the practical utility of the trade-off mentioned in contribution 1 on the Street View House Numbers (SVHN) dataset. We propose evaluating robustness in terms of *fault tolerance* by plotting the information transmission rate versus the signal-to-noise

ratio for unbounded and unseen "worst case" noise. Furthermore, we demonstrate successful communication across the DNN in the reverse direction by generating plausible messages from only the label, confirming the expectation that a communication channel should work bidirectionally.

## 2 THEORY

We begin by formulating the machine learning problem in the framework of information theory, focusing in particular on image classification, with the model being a feedforward deep neural network (DNN) trained in the supervised setting. We find that this problem has a natural interpretation as that of communication over noisy channels and use it as a concrete example for the following analysis (which is, however, not limited to image classification). We assume basic familiarity with information-theoretic quantities, such as entropy and mutual information.

The model's input (image), the corresponding desired output (true label) and the actual output (predicted label) are represented by the random variables $X, Y$ and $\hat{Y}$, respectively. The goal of the classification task is to adjust the parameters of the model such that the prediction $\hat{Y}$ given $X$ is as close as possible to the true label $Y$ for previously unseen inputs. The mapping from input $X$ to prediction $\hat{Y}$ is formally characterized through the conditional probability distribution $P(\hat{Y}|X)$. We model each of the $n_L$ layers in a DNN as a continuous communication channel, and describe the corresponding output as a random variable $T_j$, $j = 1, \ldots, n_L$, as in Tishby & Zaslavsky (2015). The DNN is the series connection of such channels. In the communication setting, the label $Y$ is the intended message, and the image $X$ can be interpreted as a highly redundant and noisy encoding of $Y$. The decoded message $\hat{Y}$ is obtained from the final layer $T_{n_L}$.

A central quantity characterizing a physical channel is its capacity, defined as the maximum transmission rate at which communication is possible with arbitrarily low probability of error. Consider that in order to transmit information over a channel, the message must first be encoded in terms of signals that the channel can actually represent – for example, voltage pulses of varying amplitude. These signals are composed to sequences of length $n$, or "codewords", used to represent the message. An ideal continuous channel allows for infinitely many codewords that are all uniquely decodable, and thus achieves error-free information transmission, corresponding to infinite capacity. In practice, channel capacity is restricted by two factors: the available power and the noise, e.g., from the environment. The most common model used in theory to characterize real channels is the Gaussian channel with additive white noise $Z \sim \mathcal{N}(0, N)$, depicted in Figure 1(a), with a constraint on the average power of the signal that is proportional to the signal amplitude $x_i^2$: $\frac{1}{n} \sum_{i=1}^{n} x_i^2 \leq P$. Shannon (1948) proved that the information capacity of the Gaussian channel is a function of the signal-to-noise ratio (SNR) $P/N$:

$$C = \frac{1}{2} \log_2 \left( 1 + \frac{P}{N} \right) \qquad \text{bits per transmission.} \qquad (1)$$

The maximal number of codewords that can be decoded with an arbitrarily small probability of error under these constraints can be estimated via a simple geometrical argument: An input sequence $\mathbf{x} = \{x_i\}_{i=1}^{n}$ and the corresponding received sequence $\mathbf{y} = \{y_i\}_{i=1}^{n} = \{(x_i + z_i)\}_{i=1}^{n}$ can be represented as two points in an $n$-dimensional space of encodings. The average power constraint forces $\mathbf{x}$ to lie inside a sphere of radius $\sqrt{nP}$ centered around the origin. Since $\mathbf{y}$ differs from $\mathbf{x}$ only due to noise with variance $N$, it will be found with high probability in a sphere of radius $\sqrt{nN}$ centered around $\mathbf{x}$. In this way, the sphere with radius $\sqrt{n(P + N)}$ is partitioned into decoding spheres with radius $\sqrt{nN}$, one for each codeword $\mathbf{x}$. The maximum number of non-confusable codewords $M$ is given by the number of non-intersecting decoding spheres that can be "packed" into the large sphere, obtained from the ratio of sphere volumes:

$$M = \left( \frac{P + N}{N} \right)^{n/2} = 2^{\frac{n}{2} \log_2 \left( 1 + \frac{P}{N} \right)}, \qquad (2)$$

where the exponent in the final expression is precisely $nC$, since capacity $C$ is the maximal achievable rate.

In image classification setting, the noise is already built into $X$, as indicated in Figure 1(b). The fact that we wish to transmit only the message, modeled as a latent cause variable $\Phi$, observable only

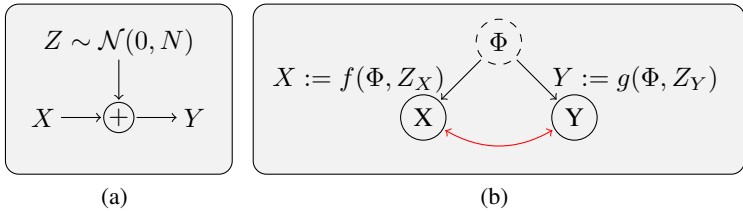

Figure 1: (a) The additive white Gaussian noise (AWGN) channel (Cover & Thomas, 1991). The additive noise $Z$, originating from various sources in the environment, is assumed to be independent of $X$. (b) Structural causal model of a pattern recognition task, with the intended message as a latent random variable $\Phi$, causal mechanisms $f$, $g$, and noise variables $Z_X$, $Z_Y$; adapted from Peters et al. (2017). The red arrow from $X$ to $Y$ represents the DNN channel. By assumption, a learnable classification task must have a causal relationship between input and output, and the ideal way to learn the rule is by uncovering this causal relationship.

*with noise* through $X$ – but not $X$ itself – is an essential detail that lets us reason about deterministic neural network layers as noisy channels, even though no noise is added in the channel itself after the input layer.[1] An average power constraint on the input to the channel for unbounded activation functions such as the relu is equivalent to a squared $L_2$ norm constraint on the weights – i.e., the traditional weight decay – since the input to a DNN is typically bounded, and the input to the channel is the weighted input of the layer.

The geometric sphere packing argument can be extended to understand the role of capacity in supervised deep learning: We say that the model has learned the rule if training inputs are mapped to $n_K$ non-overlapping clusters, corresponding to the associated labels. The model generalizes to the extent that test inputs are mapped to the same clusters. Achieving this already means a perfect solution to the classification task in the sense of arbitrarily small prediction error. However, this is not sufficient for fault-tolerance, or generalization in the stricter sense, since all vacant encoding space that is not assigned to any label represents "excess capacity". To achieve fault-tolerance, we have to minimize excess capacity, or, in terms of the geometric interpretation, we need to maximally fill out the available encoding space. The ideal result with the densest packing of a spherical volume requires the $n_K$ non-overlapping clusters to be spherical. Spherical cluster shape can be achieved by whitening (i.e., sphering) the input, and by learning projections that explain the natural variability in the data to the extent that it is relevant.

Provided a sufficient power budget, the representation learning problem is that of finding projections that rotate and reflect the input into compact volumes that can be partitioned homogeneously with respect to the label. The trade-off is that making the class-specific volumes more compact requires compression of information, but we need to preserve enough information to distribute the decoding spheres among the available representation space such that they are all non-overlapping. This illustrates how the problem of reliable communication over a noisy channel is essentially that of optimal information compression. The transmission rate $R$ is a function of the number of codewords $M$ and their length $n$: $R = \log(M)/n$. Increasing $M$ for fixed channel properties ($P$ and $N$) will increase the overlap between decoding spheres, while decreasing $n$ corresponds to compressing the input representation. Therefore, increasing $R$ is only possible at the cost of transmission accuracy. This fundamental trade-off is characterized analytically by rate-distortion (RD) theory in the form of a constrained optimization problem. The RD theory introduces a distortion measure $d(x, t)$ that quantifies the distance between a random variable $X$ and its representation $T$, and describes the rate $R$ as a function of the average distortion $D$.

A major drawback of RD theory is that it does not specify how to choose $d$. The IB method, which can be seen as a generalization of RD theory, answers this question by introducing the concept of the relevant variable $Y$. In this setting, the representation $T$ is required to maximally preserve information that $X$ carries about $Y$ (Tishby et al., 1999). This way, the constrained optimization

---

[1] We acknowledge that stochastic feedforward DNNs are considered in the literature (c.f. Tang & Salakhutdinov, 2013), but only consider the more typical deterministic layers in our work.

problem is expressed solely in terms of mutual information:

$$\mathcal{L} = I(X;T) - \beta I(Y;T), \tag{3}$$

where $I(X;T)$ measures the compression of $X$ and $\beta$ is a positive parameter associated with the constraint on the preserved information $I(Y;T)$. Thus, the solution to (3) characterizes an optimal trade-off between compressing $X$ and preserving the relevant information it carries.

It turns out that it is more efficient to split up the task of encoding the message into multiple stages via a *successive refinement* of information (Equitz & Cover, 1991), which means that we should be able to decode the relevant variable $Y$ from *any* of the layers $T_j$ with distortion $D_j$, and the description will be RD-optimal at each layer. In DNNs, each of the "stages" are the hidden layers' representations $T_j$ of the input $X$, which induce a partition of the input space w.r.t. the relevant variable $Y$ and coarsen it gradually in the course of learning. This corresponds to the case of no excess capacity.

Remarkably, it has been suggested that optimizing all layers in a DNN simultaneously with plain stochastic gradient descent (SGD) is sufficient to achieve successive refinement. The process is characterized by a brief fitting or memorization phase where $I(X;T)$ and $I(Y;T)$ increase monotonically – consistent with empirical observations that DNNs begin memorizing after the first epoch (Carlini et al., 2018) – and followed by a compression phase where $I(X;T)$ decreases.

If there are no constraints on the complexity of the system, there is no need for successive refinement: relevant information can always be encoded with sub-optimal representations at a higher cost, e.g., through some degree of memorization. Such a solution achieves small distortion (error) with significant excess rate (complexity), making it easier to transmit messages along the channel which are decoded arbitrarily at test time – i.e., adversarial examples.

It is argued that the primary mechanism facilitating compression is the inherent noise in the dataset, which increases the entropy of the weights subject to a constraint on the empirical risk. This entropy increase is logarithmic in the number of epochs, consistent with observations that we may reap substantial benefits by continuing to train well after the empirical loss is minimized, e.g., for obtaining a maximum-margin solution (Soudry et al., 2018). Although optimal compression may be conferred automatically by SGD given sufficient data, it is prudent to take other measures to improve learnability, e.g., by whitening the input, to achieve this in fewer epochs.

## 3 IMPLICATIONS

The interplay between the three fundamental quantities $n$, $N$, and $P$ in sphere packing is the key to a clear understanding of works that study the generalization performance and capacity control of DNNs. Supplementary sphere packing visualizations are provided in Appendix A.

### 3.1 CAPACITY

Bartlett (1998) showed that if a multi-layer network with sigmoidal units can be obtained with low training error, then the generalization performance depends on the size of the weights rather than the number. In the sphere packing argument, the norm of the weights affects $P$ only, and by extension, the volume of the outer sphere. The number of weights $n$ scales both the small and large sphere proportionally. Thus, assuming the case that the model can learn the rule at all, changing $P$ significantly impacts the excess capacity, wheras $n$ does not.

Neyshabur et al. (2015) showed that capacity cannot be controlled in general through $P$ alone for unbounded $n$. In a linear model, however, $n$ is set by the data and is therefore finite, so we can always bound capacity through $P$.

The experiments of Zhang et al. (2017) raise the question: Should we be surprised if a particular architecture with $n$ larger than the number of examples fits random labels? If we bound $P$ – yes, otherwise – no. We cannot establish to which extent Zhang et al. limit $P$, and thus the extent to which we would expect their models to generalize. Generalization gap in terms of accuracy is only a proxy for that in terms of the loss, which is a signal more sensitive to $P$. Furthermore, many of their architectures use batch normalization layers that introduce a learnable scaling parameter that is not penalized, and therefore the available capacity is infinite (for unbounded activation functions).

Implicit regularization by early stopping is a natural form of capacity control in that $P$ will be finite, but this addresses only the symptoms of an ill-posed learning task – not the cause.

Saxe et al. (2018) did not observe compression in terms of $I(X;T)$ for models with unbounded activation functions. The information is consistent with the available capacity being infinite in these experiments as $P$ was not bounded, e.g., with weight decay. We confirm that it is possible to observe compression without bounding $P$, but the excess capacity will not be minimized.

### 3.2 Adversarial Examples

The existence of a trade-off between a model's prediction accuracy and sensitivity to adversarial examples – where arbitrarily low error implies greater sensitivity – is not yet universally accepted.

Tanay & Griffin (2016) proposed a taxonomy of adversarial examples for linear models, suggesting a basic trade-off. They define "Type 1" examples that affect *optimal classifiers*, such that "the inconvenience of their existence is balanced by the performance gains allowed". This view has been maintained for DNNs (Galloway et al., 2018) and advanced more formally by Tsipras et al. (2018).

On the other hand, Gilmer et al. (2018) examine a situation for which *non-zero* error *implies* that a model is sensitive to small perturbations. There is a simple explanation for the apparent contradiction: The synthetic dataset considered by Gilmer et al. has no noise, so there is no rate-distortion trade-off, and the optimal strategy is to drive the error to zero. However, this is not representative of computer vision for natural images, where the intended message is always observed with noise.

Dube (2018) draws on high-dimensional geometry and attributes adversarial examples to "negative space" that is unoccupied by legitimate image manifolds; this can be interpreted as excess rate in the context of RD theory and our channel analogy. Alemi et al. (2017) make a direct connection between the IB principle and adversarial examples. They use $\beta$ in the constrained optimization (3) as regularization, but do not formalize the problem in terms of excess rate. Tuning $\beta$ selects an RD optimal trade-off, with rate $I(X;T)$ as a distortion constraint, but is not viewed as a regularizer in IB theory (Chechik et al., 2005).

Chalupka et al. (2015) suggest partitioning the information that $X$ contains about a relevance variable $Y$ into visual causes $\Phi$, and spurious correlates $S$. We attribute the most impressive adversarial examples – in the sense that they are indistinguishable to the human eye – to the model fitting spurious correlates, which are not typically removed when optimizing for high accuracy. We suggest that $\Phi$ is equivalent to robust primary features, and $S$ the weak secondary predictors, recently identified by Tsipras et al. (2018) and Tanay et al. (2018). Therefore, we would like to discard as much of $S$, while retaining as much of $\Phi$, as possible. This implies a drop in prediction accuracy for the training set, and possibly even the held-out test set. The payoff is conferred via robust generalization and fault tolerance, i.e., graceful failure, for worst-case inputs.

## 4 Experiments

The purpose of the experiments is to show how we can achieve stronger fault tolerance through careful parameterization of SGD and training to convergence. We first establish that convergence can be observed in terms of SNR in the weight updates. We then show how explicit regularization changes the optimization dynamics.

### 4.1 The Convergence of SGD via the Gradients' Signal-to-Noise Ratio

We examine the convergence properties of an over-parameterized six-layer fully-connected MLP trained on MNIST "3 versus 7" dataset with SGD. In Figure 2 we show the SNR for the stochastic gradients w.r.t. the parameters of this model. As suggested by Schwartz-Ziv & Tishby (2017), the SNR approaches a constant as training is prolonged, regardless of the choice of activation function. The number of epochs required to achieve this, however, differs dramatically: it is 1,000 for relu, 3,000 for tanh, but only 100 when BN is applied. The SNR levels in the model without BN converge to different values for each layer, whereas the curves for the layers with BN collapse.

For both nonlinearities without BN, the difference between the initial and final SNR level is smallest for the last layer (black), and largest for the earlier layers (e.g., red, cyan). Recall that the "signal"

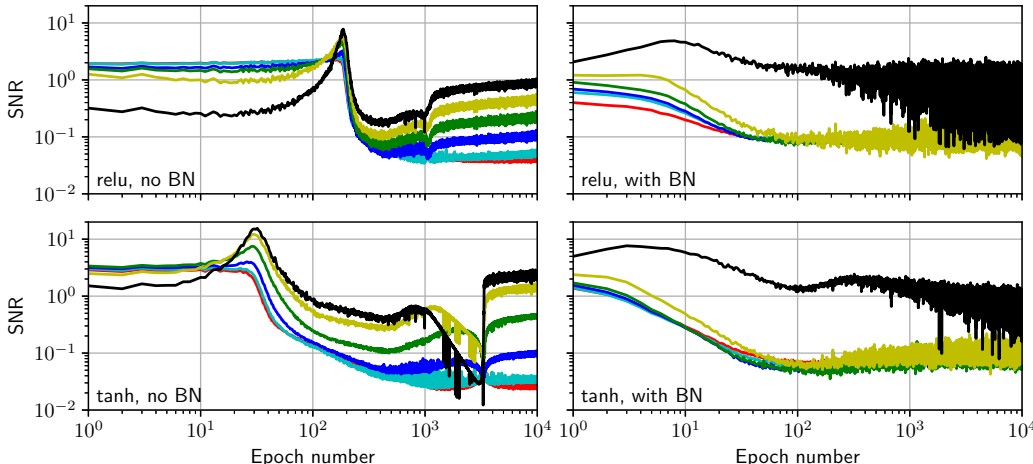

Figure 2: The SNR for the stochastic gradient of the cross-entropy loss $\mathcal{L}$ w.r.t. the weights versus epochs, with $\|\operatorname{mean}(\nabla_w \mathcal{L})\|_2 / \|w\|_2$ as signal and $\|\operatorname{std}(\nabla_w \mathcal{L})\|_2$ as noise. We use constant learning rate (1e-3) SGD with a mini-batch size of 100, relu and tanh activation functions, with and without batch normalization layers on all but the last layer. Best viewed in colour.

is a function of how much information the layer preserves about the relevant variable $Y$. At random initialization, the layers close to the input preserve all information about $Y$, while the last layer has the least. On convergence, all layers have roughly the same information about $Y$ if we neglect minor losses due to the data processing inequality, but the last layer maintains the least information about the input, and therefore has the least "noise". Thus, the trajectories in the information plane shown by Schwartz-Ziv & Tishby and in Section 4.2 of the present work, are closely related to the gradients' SNR, which predicts the sorting of the layers in descending order as observed on the left-side panels in Figure 2.

The fundamental difference between normalizing the input vs. the hidden layers, is that we only *linearly* decorrelate the input, which preserves nonlinear structure, e.g., edges, such that it is possible to spread out the decoding spheres that initially cluster around the origin and overlap. With BN, we do this normalization post nonlinearity, which is problematic for successive refinement. Furthermore, it is irrelevant if the individual neurons maintain some nonlinear effect; if the cumulative effect of the neurons in the layer is to make $T$ more normal, then this description becomes less refineable, and cannot be successively refined at all if it is exactly normal and the block length is one (Equitz & Cover, 1991). We do have a block length of one in a feedforward DNN, because we consider transmission w.r.t. one use of the channel.

## 4.2 MEMORIZATION IS THE LACK OF COMPRESSION

Arpit et al. (2017) use a working definition for memorization as "the behavior exhibited by DNNs trained on noise". Through information theory, we can provide a more quantitative definition: Memorization is the difference between the realized $I(X;T)$ – i.e., the mutual information between the output of layer $T$ and input $X$ – from the IB-optimal value. In other words, the degree of memorization is given by how much compression did not occur relative to what was theoretically possible. Importantly, this quantity has well-defined upper and lower bounds. We can estimate the lower bound $\hat{I}(X;T)$ through the data processing inequality (DPI) and plug-in a maximum likelihood estimator on samples from the training set. However, we can also upper-bound $I(X;T)$ by $H(X)$, for which a conservative estimate can always be obtained by treating the pixels as spatially independent. For example, an image with 8-bit pixels has a maximum absolute entropy of 8-bits per pixel, achieved if and only if the pixels are distributed uniformly.

Experimentally, memorization is characterized by the absence of a phase transition in the information plane, as shown in Figure 3 for original versus random labels. We can expect to see more compression for random labels in general if there is a significant amount of irrelevant information in

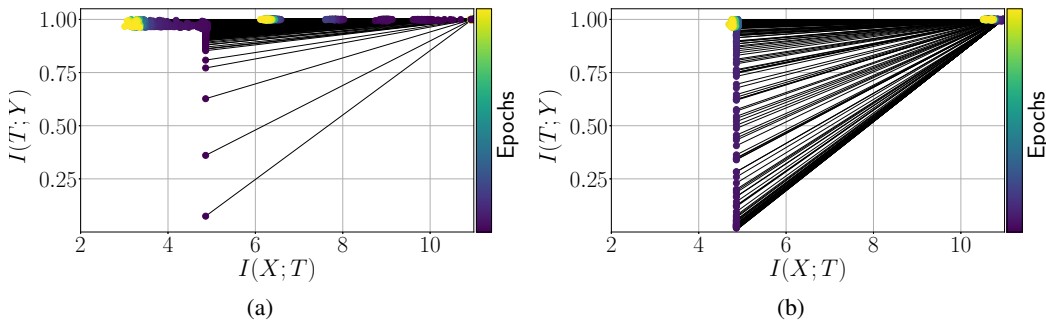

(a)             (b)

Figure 3: Information plane visualization for the model from Section 4.1 with the relu activation function trained on the original labels (a), and random labels (b) for $10^4$ epochs. Intuitively, we only see compression for the original labels, whereas the model memorizes in the random label case. The degree of memorization is roughly the difference in $I(X;T)$ values between the two cases.

the dataset, but this is not the case for MNIST 3 versus 7 in Figure 3(b). In this case, memorization is characterized by a slow monotonic increase in $I(T;Y)$ with almost no compression; however, the fact that this is possible at all implies excess rate. As the relu activation function is not bounded, we estimate a lower bound on the information with a maximum entropy adaptive binning scheme that bins the CDF for each layer equally by recomputing the bin ranges every epoch. We use 30 bins and 2,000 samples from the training set. See Darbellay & Vajda (1999) and Paninski (2003) for an overview of such methods.

### 4.3 EXPLICIT REGULARIZATION AND THE GENERALIZATION GAP

In the next two sections we focus on the Street View House Numbers (SVHN) dataset (Netzer et al., 2011), which is an appropriate dataset for characterizing adversarial robustness now that compelling results have been shown for 3 versus 7 (Tanay & Griffin, 2016) and full MNIST (Schott et al., 2018). Learning the SVHN dataset can be considered more "difficult" than MNIST due to having significantly more noise, e.g., from the "distracting" digits in addition to the relevant one, and occupying a larger RGB canvas in $R^{32 \times 32 \times 3}$. Our primary objective is not to construct an "adversarial defense", but simply to maximize out-of-sample performance, i.e, generalize to the global population of house numbers using Arabic numerals. We characterize the fault tolerance of our models for unbounded adversarial examples in Section 4.4.

Intuitively, colour and texture are not legitimate causes of a digit's class, so we first convert from RGB to grayscale (NTSC). We also apply PCA, retaining the top 400 principal components, and then a zero-phase whitening (ZCA) to linearly decorrelate neighboring pixels (Bell & Sejnowski, 1997). This increases the SNR by emphasizing the information in the edges.

We characterize two explicit regularization strategies applied to a simple four-layer CNN (see Appendix B for the architecture): batch normalization (Ioffe & Szegedy, 2015) without momentum, applied to the two middle hidden layers, and squared-$L_2$ weight decay. The training regime was inspired by the IB method applied to deep learning (Schwartz-Ziv & Tishby, 2017). We parameterize SGD's stationary distribution with deliberately chosen hyperparameters; no automated search was invoked, then run the system to steady state.

We trained all models for 500 epochs, as this was at least one order of magnitude longer than the "fitting" phase required to minimize the training loss. This would allow sufficient time to observe compression. We chose a relatively large and constant learning rate of 1e-2 to maximize the power of the noise during the compression phase, and by extension the energy used to maximize the entropy of the weights under the ERM constraint in the finite number of epochs. Lastly, we used the smallest weight decay regularization constant $\lambda$, i.e., the average power constraint, that prevented the model from fitting random labels significantly better than chance ($0.20 \pm 0.03\%$ absolute percent), as shown for Model B in Table 2. This $\lambda$ turned out to be approximately 1e-2. Interestingly, the model with this setting of weight decay *happened* to also have the highest test accuracy.

Table 1: Test accuracy (Test (%)) and generalization gap (Gap (%)) for A: the baseline, B: A + weight decay (WD), and C: B + batch normalization (BN). There is no large-batch generalization gap (LBGG) for B; however, a small LBGG appears when we introduce BN in C. Not shown: The loss gap is one order of magnitude less for B vs. A. Training accuracy is omitted for brevity. Model A achieves $100\pm0\%$ training accuracy. We report the mean accuracy and standard error of the mean, assuming the error is normally distributed and independent over 5 seeds.

| | A | | B | | C | |
|---|---|---|---|---|---|---|
| Batch Size | Test (%) | Gap (%) | Test (%) | Gap (%) | Test (%) | Gap (%) |
| 128 | $85.9 \pm 0.2$ | $14.1 \pm 0.2$ | $88.4 \pm 0.2$ | $4.8 \pm 0.1$ | $87.1 \pm 0.4$ | $12.42 \pm 0.04$ |
| 64 | $86.3 \pm 0.1$ | $13.7 \pm 0.1$ | $87.6 \pm 0.1$ | $5.5 \pm 0.2$ | $85.1 \pm 0.5$ | $11.3 \pm 0.2$ |
| 32 | $86.6 \pm 0.1$ | $13.4 \pm 0.1$ | $87.2 \pm 0.4$ | $5.4 \pm 0.2$ | $85.2 \pm 0.4$ | $9.0 \pm 0.3$ |

Table 2: Models A, B, and C from Table 1, but trained instead with random labels sampled uniformly. Clearly, model B has the least information capacity, yet also happens to achieve the highest test accuracy for the original labels in Table 1. Notice the $25\pm1\%$ LBGG for C, and the $54\pm2\%$ GG between C and B for batch size 128. There is a $3 \pm 4\%$ LBGG in A.

| | A | | B | | C | |
|---|---|---|---|---|---|---|
| Batch Size | Train (%) | Gap (%) | Train (%) | Gap (%) | Train (%) | Gap (%) |
| 128 | $85 \pm 3$ | $76 \pm 3$ | $10.20 \pm 0.03$ | $3 \pm 2$ | $67 \pm 1$ | $57 \pm 1$ |
| 32 | $90 \pm 2$ | $79 \pm 2$ | – | – | $42 \pm 1$ | $32 \pm 1$ |

We vary the batch size, optionally adding the explicit regularizers, and report the resulting generalization gap in Table 1.[2] Model C includes weight decay and batch normalization because the baseline model A already perfectly fits the training set, adding batch norm alone yields a similarly large generalization gap as the baseline. Results for the same experiments repeated with random labels are provided in Table 2, where we demonstrate that Model B (baseline + weight decay) has a generalization gap of $3 \pm 2\%$. This particular generalization gap is not to be trusted, since the SVHN test set is unbalanced. In fact, one particular seed achieved a generalization gap of 9.4%, or a test accuracy of 19.6%, which is exactly the accuracy obtained by always predicting class 1. The information is 0 bits in this case, but we report accuracy here in keeping with standard practice.

Using the strategy that minimized generalization error (Model B), we retrain with additional training data to boost the test accuracy (Model B+), gaining $2\pm0.3\%$ for a $0.3\pm0.1\%$ smaller generalization gap. An additional generalization result to handwritten digits can be found in Appendix C.

### 4.4 Fault Tolerance

We demonstrate bidirectional communication with model B+ in Figure 4. The human legible outputs shown in Figure 4(b) were obtained via a similar gradient-based procedure used by Nguyen et al. (2015) to craft "fooling images" initialized from noise. There was little qualitative difference to these samples compared to those obtained from Model B trained on only the smaller training set consisting of images with distracting digits, which we show in Figure 4(a) along with their prediction confidences.

In Figure 5 we characterize fault tolerance in terms of information conveyed about the label that gradually decreases with decreasing SNR. We compute SNR in dB as in Cisse et al. (2017) for inputs $x$ from the test set, and noise $\delta_x$ as $20 \log_{10} \left(1 + \frac{\|x\|}{\|\delta_x\|}\right)$ for three noise variants. The first two are adversarial and obtained by iteratively minimizing the loss w.r.t. the input with a step size of 1e-2, then either taking the real $L_2$-normalized gradient or the signed gradient, denoted as $L_2$ and $L_\infty$, respectively, in Figure 5. As baseline we compare with white Gaussian noise.

The model trained with batch normalization conveys less information for all sources of noise at the same SNR than without. Note that the information plateau in Figure 5(a) after SNR of 30dB for the adversarial noise is not due to the gradient masking effect – the accuracy indeed goes to 0% for this unbounded attack. At this point, there is so much salient structure in the noise that we are

---

[2] The significant figures are consistent with the precision of the standard error in each experiment.

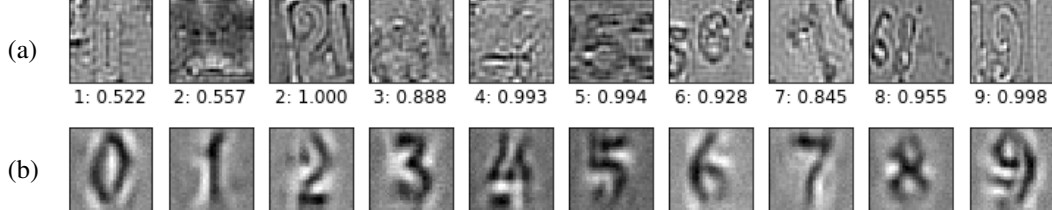

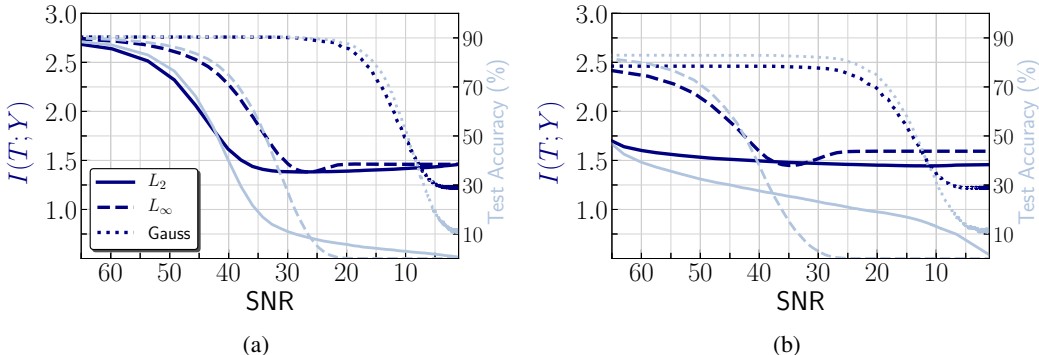

Figure 4: Demonstration of regularized generative modeling via an adversarial attack. (a) Samples from our preprocessed version of the SVHN dataset for each class $Y \in \{0, \dots, 9\}$ arranged from left to right, with the predicted class (argmax softmax probability) and confidence. (b) Transmitting a message across the channel in the reverse direction (i.e., from $Y$ to $X$) by minimizing the loss for each $y \in Y$ w.r.t. $X \sim \mathcal{N}(\mu, \sigma^2)$ with step size 1e-2. We set $\mu$ equal to the population mean, and $\sigma^2$ as two orders of magnitude less than the population standard deviation. We show the pattern $X$ obtained after iterating until full confidence to within 3 decimal places, which took $\approx 100$ steps. Unlike in Nguyen et al. (2015), we recover prototypical examples because we have reduced the excess rate of the channel. This would not be a suitable model of the original data-generating distribution as it is implicitly regularized by $Y$ such that the irrelevant digits in (a) are forgotten.

Figure 5: Characterizing fault tolerance in terms of information transmitted, $I(T; Y)$ for 10K samples from the SVHN test set, as a function of the SNR for two kinds of adversarial noise ("$L_2$" and "$L_\infty$"), and white Gaussian noise ("Gauss"). Model (a) is regularized with weight decay, while (b) uses BN and is less tolerant to all sources of noise. Accuracy is eventually reduced to $0\%$ for the unbounded attacks and $\approx 10\%$ for Gaussian noise, however the images with model dependent adversarial "noise" contain more information (as expected) as the SNR $\to 0$.

interpolating between legitimate images. The information slowly rises as the model reliably predicts a specific class as if it were a targeted attack, even though the original goal was only to make the prediction `not` $Y$. This conveys some information about $Y$ since there are several alternatives that are consistently not chosen, e.g., for large perturbations '9's are mapped to '6's more often than not. The plateau for the less robust model in Figure 5(b) can be explained given that $L_\infty$ perturbations induce significantly more noise in pixel space for the same decrease in information about the label, therefore the attack cannot follow the least Euclidean distance to the nearest misclassified volume.

## 5 CONCLUSION

The established framework for characterizing information transmission in the presence of noise is Shannon's rate-distortion theory. The insight that essentially the same trade-off is central to deep learning problems allows us to establish a notion of capacity for DNNs that explains their generalization behaviour to a significant extent. Guided by the IB principle, which augments rate-distortion theory with the relevant variable, and minimizing the difference between the empirical and expected loss – as suggested by statistical learning theory – we derive guidelines for efficient use of this ca-

pacity and obtain a recipe that yields compelling fault tolerance for "worst-case" inputs, such as adversarial examples. We confirm that it is indeed possible to generalize in a narrow sense to the "clean" test set when the model has excess capacity, but that minimizing this excess capacity is essential for fault-tolerant generalization behaviour.

The implications for practical deep learning are simple: To obtain a more fault tolerant model, i) irrelevant information in the dataset, such as, e.g., colour and texture in the task of digit recognition, can be safely removed; ii) training with constant learning rate SGD should be prolonged well beyond the number of epochs required to minimize the empirical loss; iii) sufficient constraints, e.g., weight decay, have to be applied. Particular care has to be taken when applying batch normalization, since it impedes successive refinement of information in the learning procedure and may harm robustness. As a general rule, we suggest that preventing the model from fitting random labels better than chance is a good first step for calibrating such constraints.

Future work will consider the role of the DNN architecture in more detail – which was set aside to emphasize general properties of SGD and the explicit regularizers – for obtaining fault tolerance to a broader array of challenging inputs such as adversarial deformations. The interpretable fooling images, and fault tolerance for unbounded $L_p$ perturbations obtained in the present work, without having optimized these metrics directly, i.e. with adversarial training, are important steps to this end.

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

# A    SPHERE PACKING

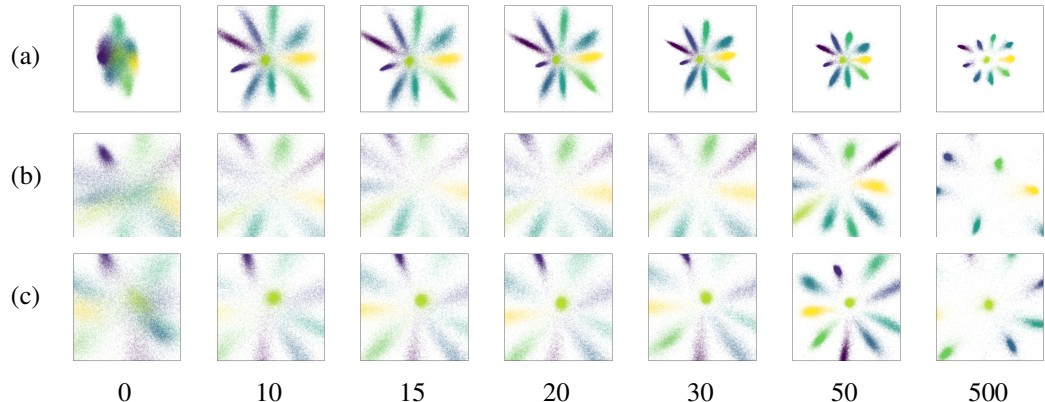

|     | 0 | 10 | 15 | 20 | 30 | 50 | 500 |

Figure 6: Sphere packing in a two-neuron ($n = 2$) layer over 500 epochs. Architecture is a fully-connected MLP [(a) (784–392–196–2–49–10) and (b) (784–392–BN–196–BN–2–49–10)] with relu units, batch size 128, constant learning rate 1e-2, weight decay $\lambda$=1e-3, and $\beta$=5e-2. All other settings left as default from Kolchinsky et al. (2017). The plots have a fixed x- and y-axis range of $\pm 10$. All samples from the MNIST training set are plotted and colour coded by label.

We use the source release from Kolchinsky et al. (2017) which exposes the IB Lagrange multiplier, $\beta$. We adapt their architecture which has a 2D layer perfect for observing the sphere packing effect with no arbitrary dimensionality reduction, e.g., by PCA or t-SNE.

Figure 6(a) depicts how the flow of information about the relevant variable is impeded immediately following random initialization, as there is no label homogeneous partition. The data are spread further apart over the first $\approx 20$ epochs, with little emphasis on compression. Once it is possible to partition the clusters, a phase transition begins in which the clusters become more circular and power is simultaneously reduced. Note how it would be possible to trivially observe circular clusters without compression by arbitrarily increasing the inter-cluster distance with more power, thus hiding the noise, but this is suboptimal and implies excess capacity.

Figure 6(b) introduces two batch normalization layers with default settings before the 2D layer (Ioffe & Szegedy, 2015). There are notable differences compared to Figure 6(a): i) it is easier to partition the clusters immediately after initialization as the data are more spread out, contributing to a "short circuit" of information between the input and output along the Markov chain, ii) the clusters are more stationary and there is no clear phase transition, iii) there is more excess capacity; the inter-cluster distance, as well as the clusters themselves, are larger. In Figure 6(c) we disable the scaling parameter $\gamma$ to isolate the effect of the normalization itself on the excess capacity. The situation improves slightly as the arrangement of the clusters is more circular, but the excess capacity remains because the normalization has made it more difficult to achieve a successive refinement of information (Equitz & Cover, 1991).

# B    MODEL ARCHITECTURE

We describe the CNN architecture used for the SVHN experiments in Table 3, which was adapted from the CleverHans library tutorials (Papernot et al., 2018).

# C    ZERO AND ONE-SHOT TRANSFER

We were curious to see if the best SVHN model would generalize to handwritten digits. To this end, we bilinearly upsample MNIST to a $32 \times 32$ grid, but do not apply any preprocessing other than to normalize digits in the range [0, 1]. The model obtained 72.6% accuracy zero-shot. Randomly selecting one example from the MNIST validation set for each class, and fine-tuning with SGD for ten steps while rotating the ten instances randomly $\leq 10°$ at each step, boosted overall test accuracy

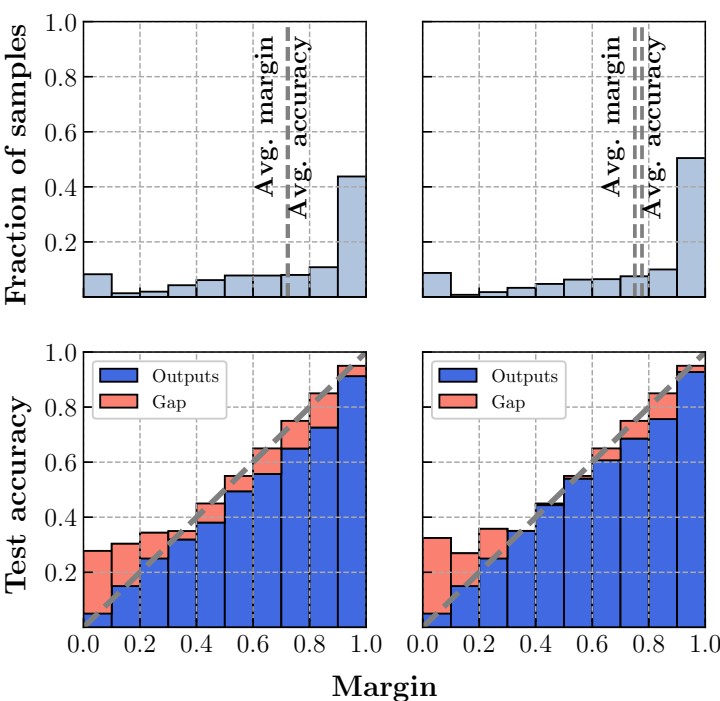

Figure 7: The model with prolonged weight decay training on SVHN (Model B+) has well calibrated predictions in a transfer learning setup to MNIST. (*Top*) Histogram of prediction margins for zero-shot (*left*), and one-shot (*right*) transfer. (*Bottom*) Test accuracy versus binned prediction margins. Although the accuracy on the full test set lags what can be obtained by training on MNIST, the model is still useful in this setting as the confidence is well calibrated despite the distributional shift. Presence of "Gap" above the diagonal line indicates under-confidence, while the same below the line means over-confidence.

Table 3: Basic fully-convolutional CNN architecture. We respectively denote $h$, $w$, $c_{in}$, $c_{out}$, $s$ as convolution kernel height, width, number of input and output channels w.r.t. each layer, and stride. The model uses the ReLU activation function, and a Gaussian parameter initialization scheme.

| Block | $h$ | $w$ | $c_{in}$ | $c_{out}$ | $s$ | params |
|-------|-----|-----|----------|-----------|-----|--------|
| Conv1 | 8 | 8 | 1 | 32 | 2 | 2.0k |
| Conv2 | 6 | 6 | 32 | 64 | 2 | 73.7k |
| Conv3 | 5 | 5 | 64 | 64 | 1 | 102.4k |
| Fc | 1 | 1 | 256 | 10 | 1 | 2.6k |
| Total | – | – | – | – | – | **180.9k** |

to 77.6%. This is better than figures reported elsewhere for one-shot learning, e.g., Vinyals et al. (2016) obtain 72% (they did not report a zero-shot figure), and transfer from the Omniglot dataset, which has a flat background like MNIST, rather than SVHN.

More interestingly, Figure 7 shows that the predictions are reasonably well calibrated out of the box, i.e., without temperature scaling.[3] As the model *must* provide a score for each of the ten categories, i.e., it has no explicit abstain mechanism, we use the prediction *margin*: the difference between the maximum softmax probability and second highest probability, as its confidence. Taking the probability associated with the predicted class, as in Guo et al. (2017), is not representative of the actual confidence, e.g., the prediction: 55% class 7, 40% class 1, should be interpreted as 15% confidence in class 7, not 55%. Surprisingly, the average confidence was exactly matched to the average test accuracy in the zero-shot case, which is why "Avg. margin" and "Avg. accuracy" are overlapping in the top left plot of Figure 7. Fine tuning with a single instance from each class primarily reduces over-confidence, while leaving the model under-confident for small margins, which slightly increases the gap between the average margin and the higher accuracy.

---

[3]Temperature scaling renders an analysis of the prediction margins meaningless with respect to generalization, and may cause gradient masking (Carlini & Wagner, 2016).

