# OpenReview forum: "A Rate-Distortion Theory of Adversarial Examples"
_ICLR.cc/2019/Conference_

### Official Review · AnonReviewer3 · 2018-11-03
**In my opinion the paper is so poorly written that it makes it difficult for me to judge it**

**Rating:** 2
**Confidence:** 3

**Review:**

This paper tries to draw connections between rate distortion theory and DNNs and use some intuitions from that domain to draw conclusions about robustness and generalization of the DNNs.

The paper is mostly written in a storytelling narrative with very little rigor. In my opinion, this lack of rigor is problematic for a conference paper that has to be concise and rigorous. Moreover, the story is not told in a cohesive way. In most parts of the paper, there is not much relationship between the consecutive paragraphs. And even within most of the paragraphs, I was lost in understanding what the authors meant. I wish the paper would have been self-contained and made concrete definitions and statements instead of very high-level ideas that are difficult to judge. In the current state, it is very difficult for me to say what exactly is the contribution of the paper in terms of the story other than some loosely related high-level ideas. I feel like most parts the story that the authors are telling is already told by many other papers in other forms(papers that authors have cited and many other ones).

---

### Official Review · AnonReviewer1 · 2018-11-03
**While the experiment of fault tolerance is interesting, the obtained implications are somewhat trivial.**

**Rating:** 3
**Confidence:** 3

**Review:**

This paper considers the trade-off between the prediction accuracy of deep neural networks (DNNs) and sensitivity to adversarial examples.
Reviewing the (Gaussian) channel capacity and rate-distortion theory, i.e., the information bottleneck, the authors discuss their implications on the generalization performance of DNNs. The experiments demonstrate the SNR of gradients, information plane, the generalization gap, and fault tolerance against adversarial examples.

While the interpretations of DNN learning by the information theoretic concepts are interesting, most of them are already known results, and hence provide little novel theoretical knowledge.

The discussions in Section 3 are superficial. It is not clear how they are related to the main arguments of this paper.

While the experiment of fault tolerance is interesting, the implications obtained from experiments are somewhat trivial.

minor comments:
p.2, l.15: h, w, and c are undefined.
Section 2: Rate-distortion theory is usually explained by the sphere covering argument instead of sphere packing.
Section 4.3.1: It is not explained what zero and one-shot transfer learning is.

Pros:
The experiment of fault tolerance is interesting.
Cons:
Theoretical parts are basic results of information theory.
The implications of experiments are somewhat trivial.

---

### Official Review · AnonReviewer2 · 2018-11-05
**see below**

**Rating:** 4
**Confidence:** 4

**Review:**

The paper discusses on a rate distortion interpretation of adversarial examples by building the equivalence of DNN and a noisy channel. The proposed topic is very interesting. However, it is quite disappointing after reading the paper, that it does not deliver. In a sense, the reader has an impression that the paper is a collection of fractions of small thoughts and empirical observation pieces that are yet to be stringed up coherently.
*To start with, the contributions are not clear. The major equations (1-4) are all pre-existing. The main Figure (fig.1) is also not new. Sec.3 is on implications, while it is more a discussion section centered on existing works about capacity and adversarial examples. Although it is claimed in the beginning of the paper 3 theoretical and empirical contributions, they are not clearly presented in the follow-up text.
*Empirical evaluation, in Fig.2, a legend should be in place to introduce the colored curves. Currently it is unclear what it is for each of the curves.
*Fig.3: it is unclear why the MI plots are of piecewise straight lines. Does it imply that the two MIs are linearly related?
*Table 1&2: Not clear how this observation has to do with the RD theory.

Seems no response from the authors.

---

### Author Response · Authors · 2018-12-06
**Response to reviewers**

We thank the reviewers for their honest feedback. We agree that the ideas were not presented clearly and will work on improving this.

---

### Meta-Review · Area_Chair1 · 2018-12-15
**needs better presentation**

**Confidence:** 5
**Recommendation:** Reject

**Metareview:**

Both authors and reviewers agree that the ideas in the paper were not presented clearly enough.